# Demonstration of valley anisotropy utilized to enhance the thermoelectric power factor

Airan Li [1,3], Chaoliang Hu [1,3], Bin He[2], Mengyu Yao [2], Chenguang Fu [1✉], Yuechu Wang [1], Xinbing Zhao [1], Claudia Felser [2] & Tiejun Zhu [1✉]

Valley anisotropy is a favorable electronic structure feature that could be utilized for good thermoelectric performance. Here, taking advantage of the single anisotropic Fermi pocket in p-type $Mg_3Sb_2$, a feasible strategy utilizing the valley anisotropy to enhance the thermo-electric power factor is demonstrated by synergistic studies on both single crystals and textured polycrystalline samples. Compared to the heavy-band direction, a higher carrier mobility by a factor of 3 is observed along the light-band direction, while the Seebeck coefficient remains similar. Together with lower lattice thermal conductivity, an increased room-temperature $zT$ by a factor of 3.6 is found. Moreover, the first-principles calculations of 66 isostructural Zintl phase compounds are conducted and 9 of them are screened out displaying a $p_z$-orbital-dominated valence band, similar to $Mg_3Sb_2$. In this work, we experi-mentally demonstrate that valley anisotropy is an effective strategy for the enhancement of thermoelectric performance in materials with anisotropic Fermi pockets.

[1] State Key Laboratory of Silicon Materials, School of Materials Science and Engineering, Zhejiang University, Hangzhou, China. [2] Max Planck Institute for Chemical Physics of Solids, Dresden, Germany. [3] These authors contributed equally: Airan Li, Chaoliang Hu. ✉email: chenguang_fu@zju.edu.cn; zhutj@zju.edu.cn

Thermoelectric (TE) devices, which convert heat into electricity or vice versa, exhibit promising applications in aerospace explorers as power generators, room-temperature refrigeration, and energy supply for the Internet of Things[1], etc. The conversion efficiency is governed by the dimensionless figure of merit $zT$ of the used TE materials, $zT = S^2\sigma/(\kappa_L + \kappa_e)$, where $S$, $\sigma$, $\kappa_L$, and $\kappa_e$ stand for Seebeck coefficient, electrical conductivity, lattice and electronic component of the thermal conductivity $\kappa$, respectively[2]. The challenge in the enhancement of $zT$ lies in that these transport parameters are not independent but coupled to one another. Among them, both the $S$ and $\sigma$ are dominated by the behavior of electrons near the Fermi level $E_F$. Increasing $\sigma$ usually reduces absolute $S$. Therefore, how to decouple $S$ and $\sigma$ to realize a largely increased $\sigma$ without significantly lowering the absolute $S$ is crucial for the enhancement of the electrical power factor (PF, $PF = S^2\sigma$) in a TE material[3].

Electronic band engineering has been the leading strategy that enhances the PF of TE materials. The milestone work by Hicks and Dresselhaus proposed that the enhancement of TE performance is possible if the electrons are confined in one or two dimensions, corresponding to an abrupt change of the electronic density of states (DOS) at some energy levels[4,5], as schematically shown in Fig. 1a. This low-dimensionality strategy sparked the studies of TE materials with nanostructures, such as quantum-well superlattice structures[4], nanowires[6]. However, owing to the difficulty in large-scale synthesis and measurements of superlattice structures and nanowires, more attention, later on, has been shifted to the nanostructured bulk materials with the band engineering strategy as a powerful tool to tune the PF.

In 2008, an enhanced $zT$ of above 1.5 at 773 K in the thallium-doped PbTe was reported by Heremans et al. [7] by distorting the DOS. As illustrated in Fig. 1b, the resonant Tl-impurity level locates in the electronic band and leads to the increased DOS. Apart from distorting the DOS with the resonant level, converging the electronic bands to achieve high valley degeneracy $N_v$ was also demonstrated as a general strategy to boost the TE performance of bulk materials, such as $PbTe_{1-x}Se_x$[8] and $Mg_2Si_{1-x}Sn_x$[9]. The DOS effective mass $m_d^*$ is related to the valley degeneracy $N_v$ and single-band effective mass $m_b^*$ through the expression[10]: $m_d^* = N_v^{2/3} m_b^*$. As shown in Fig. 1c, when more bands converge to the $E_F$, a larger $m_d^*$ is produced due to the increased $N_v$, contributing to a higher DOS and $S$ without significantly reducing the carrier mobility $\mu$ if intervalley scattering is not serious[11]. From the view of band structure engineering, both resonant level and band convergence strategies target the enhanced DOS near $E_F$, leading to the improvement of $S$, similar to the low-dimensionality strategy.

Besides the DOS, the curvature of an electronic band, which is inversely proportional to $m_b^*$, is another important feature of the electronic structure that can be engineered to modulate $\mu$. A light band could guarantee a higher $\mu$ compared with the heavy one (Fig. 1d). In heavy-band TE materials, such as half-Heusler compound $V_{1-x}Nb_xFeSb$, the $m_b^*$ can be reduced by increasing Nb content, which is beneficial for a higher $\mu$ without significantly

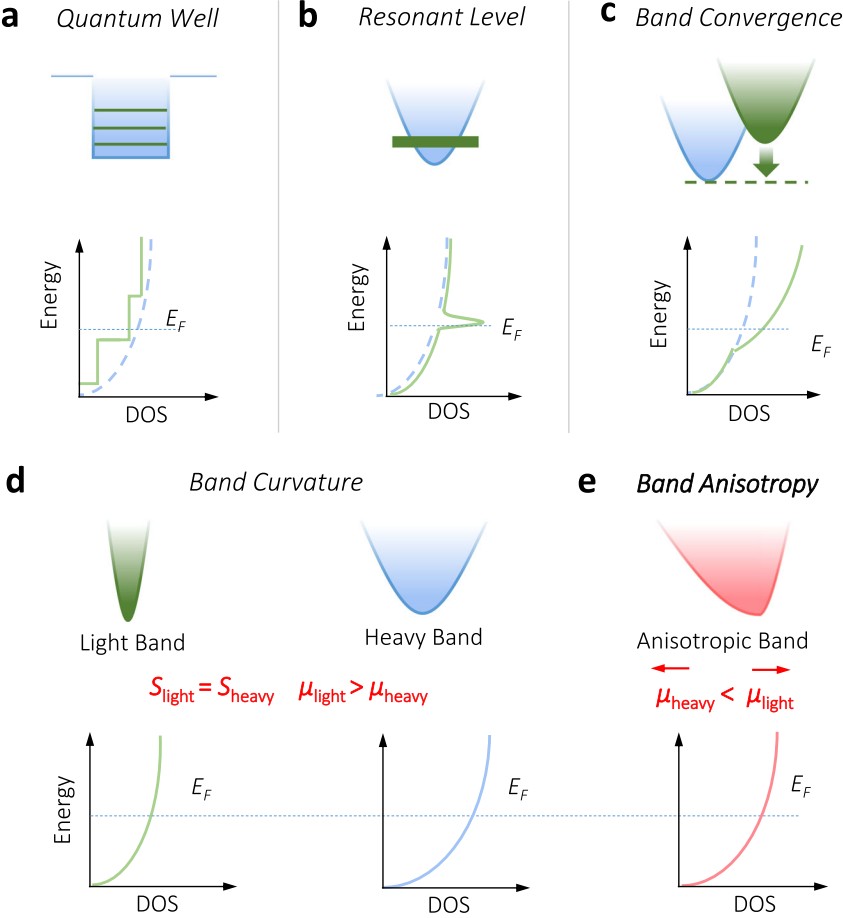

**Fig. 1 Schematic representations of the band structures and DOS for different band engineering strategies. a** Quantum well. **b** Resonant level. **c** Band convergence. **d** Band curvature and **e** band anisotropy. In **d** and **e**, the position $E_F$ is assumed to be the same for the light band, heavy band, and anisotropic band, indicating the same $S$. The difference in the $\mu$ of light- and heavy-bands and along different directions of the anisotropic band is indicated.

lowering $S$, leading to an improved PF[12,13]. The anisotropy of the electronic band along different directions is another important feature of the electronic structure in solids. Here, a band with very large anisotropy refers to a weak dispersion in one direction but a strong dispersion along the orthogonal direction[14–16] (Fig. 1e). The valley anisotropy can be gauged by the anisotropy factor, $K = m_{\parallel}^* / m_{\perp}^*$, where $m_{\parallel}^*$ and $m_{\perp}^*$ is the band effective mass along with the parallel and perpendicular directions, respectively. Combined with $N_v$, the Fermi surface complexity factor has been recently proposed as an indicator in the high-throughput search of promising TE materials[17]. The important role of valley anisotropy in enhancing the TE performance has been highlighted from the view of theoretical calculations[3,17]. However, the experimental route to utilize valley anisotropy to enhance the TE performance remains elusive.

For most heavily doped TE materials, the electronic transport properties could be understood using the single parabolic band (SPB) model (Supplementary Note 1)[18]. Assuming the carrier transport is the acoustic phonon scattering dominated, $S$ can be expressed as

$$S = k_B/e[2F_1(\eta)/F_0(\eta) - \eta], \quad (1)$$

where $k_B$ is the Boltzmann constant, $e$ is the elemental charge, $F_i(\eta)$ is the Fermi integral, $\eta = E_F/k_B T$ is the reduced Fermi level. This expression indicates that $S$ is only determined by $\eta$ (the position of $E_F$), no matter the band is light or heavy (Fig. 1d). Bearing this in mind, one can deduce that for an anisotropic band the carrier transport along the light-band direction will show a higher $\mu$ while remaining the same $S$ compared to the heavy-band direction. This suggests a feasible strategy to utilize the valley anisotropy to enhance the TE properties, specifically the PF.

Here, we experimentally demonstrate the effectiveness of utilizing valley anisotropy to enhance the TE performance by taking $p$-type Zintl phase compound $Mg_3Sb_2$ as a paradigm, which has a single anisotropic hole pocket at the center of the Brillouin zone (BZ). Based on the as-grown single crystals and textured polycrystalline samples, various methods were used to reveal the effect of valley anisotropy on the TE transport. An enhanced room-temperature $zT$ by a factor of 3.6 was found if the carriers in p-type $Mg_3Sb_2$ transport along the light-band direction, compared to the heavy-band direction. Furthermore, given the $p_z$ orbital overlapping as the indicator of valley anisotropy, the electronic structures of 66 isostructural Zintl phase compounds were calculated and 9 of them were screened out as promising candidates with potentially higher TE performance along the light-band direction.

## Results

**Valence band anisotropy in $Mg_3Sb_2$.** Experimentally, to demonstrate how valley anisotropy affects TE transport properties, it is paramount to select a suitable material system for carrying out the research. To minimize the effect of multiple pockets and the intervalley scattering, a material with a single Fermi pocket is preferable. To facilitate the analysis of the relation between the valley direction in the reciprocal space and the crystallographic direction in the real space, an anisotropic Fermi pocket locates at the center of the BZ is desirable. Moreover, the growth of single crystals should be feasible to enable the study of the anisotropic transport properties. With these criteria, p-type $Mg_3Sb_2$ single crystal[19], which has a single anisotropic hole pocket locating at the center of the BZ, was selected for this study.

The Zintl phase compounds $Mg_3Sb_{2-x}Bi_x$ have recently attracted considerable attention due to the high n-type TE performance[20–28]. Both $Mg_3Sb_2$ and $Mg_3Bi_2$ crystallize in a trigonal layered structure with the space group $P\bar{3}m1$, as shown in

Fig. 2a. Previous experimental studies[29] and calculations[30] suggest that the electron (n-type) transport of $Mg_3Sb_{2-x}Bi_x$ exhibit a very weak anisotropy, owing to the six nearly spherical electron pockets. In contrast, there is only one-hole pocket in the center of the BZ for $Mg_3Sb_2$ if the hole carrier density $n$ is below $10^{20}$ cm$^{-3}$ (Fig. 2b). The plate-like Fermi surface suggests that the hole carrier transport in the $k_x - k_y$ plane is nearly isotropic but becomes quite different along the $k_z$ direction, corresponding to the ab-plane and c-axis in the real space, respectively.

The anisotropy of $Mg_3Sb_2$ is more clearly presented in the calculated valence band structure (Fig. 2c), i.e., a large difference in the curvature along with Γ–K and Γ–A directions, which can be quantified by the effective mass: it is $0.9m_e$ along Γ–K direction while $0.11m_e$ along Γ–A direction. A $K$ value of about 8.2 is thus derived. This result is close to the calculations by Zhang et al. [31] ($m_{\Gamma-K}^* = 1.15m_e$, $m_{\Gamma-A}^* = 0.15m_e$) and Meng et al. [30] ($m_{\Gamma-K}^* = 0.61m_e$, $m_{\Gamma-A}^* = 0.07m_e$), giving the $K$ values of 7.7 and 8.7, respectively. To confirm the valence band anisotropy, we further performed angle-resolved photoemission spectroscopy (ARPES) study to reveal the experimental valence band structure of $Mg_3Sb_2$. As shown in Fig. 2d, the band curvature along Γ−K and Γ−M directions are identical, suggesting an isotropic transport along the ab-plane. In contrast, a significant steep band structure is observed along the Γ−A direction. Based on the ARPES results, the $m_{\Gamma-K}^*$ and $m_{\Gamma-A}^*$ are estimated to be $0.9m_e$ and $0.16m_e$, respectively, giving a $K$ value of about 5.6, which is close to the calculated values ($K \sim 8$). Thus, the single anisotropic valence band of $Mg_3Sb_2$ makes it an ideal system to demonstrate the effect of valley anisotropy on the TE properties.

**P-type $Mg_3Sb_2$ single crystals.** To study the effect of valley anisotropy on TE transport, high-quality single crystals with a sizable dimension are required. Previously, $Mg_3Sb_2$ single crystals were successfully grown using the flux method[19,26,27], which exhibits a thin-layered shape with a typical thickness smaller than 1 mm. Such a small thickness makes it difficult to study the TE transport properties along the c-axis. Here, the single crystals of Ag-doped $Mg_3Sb_2$ were prepared through the slow-cooling method. Silver was previously found to be a good acceptor[32] and thus used to shift the $E_F$ into the valence band. As shown in the inset of Fig. 3a and Supplementary Fig. 1a, the as-grown crystal shows clear cleavage surfaces, of which the crystallinity and orientation were checked using the XRD and Laue diffraction (Supplementary Fig. 1). The EDS mapping and line scanning were carried out to check the chemical homogeneity (Supplementary Fig. 2). The results indicate that the chemical composition is homogeneous. Bar-shaped crystals with the size of about $1 \times 1 \times 3$ mm$^3$ were cut for electrical and thermal transport measurements in the temperature range of 100–300 K.

The Hall carrier mobility $\mu_H$ of the crystals is shown in Fig. 3a. Along the c-axis, the $\mu_H$ is about 300% of that along the ab-plane in the whole temperature range, as expected from the anisotropic valence band (Fig. 2), which is in agreement with our previous study on the isostructural p-type $Mg_3Bi_2$ single-crystal reporting that $\sigma$ along the c-axis is 200% higher than that along the ab-plane[19]. These results suggest that the carrier transport along the light-band direction is faster than that along the heavy-band direction, probably owing to the weaker carrier scattering and low inertial effective mass. The $S$ and the Hall carrier density $n_H$ along both directions are shown in Fig. 3b. Very interestingly, distinct from the huge difference in the $\mu_H$, the $S$ values along both directions are close to each other, given the measurement uncertainty. These results suggest a much higher PF when the hole carriers in $Mg_3Sb_2$ move along the c-axis.

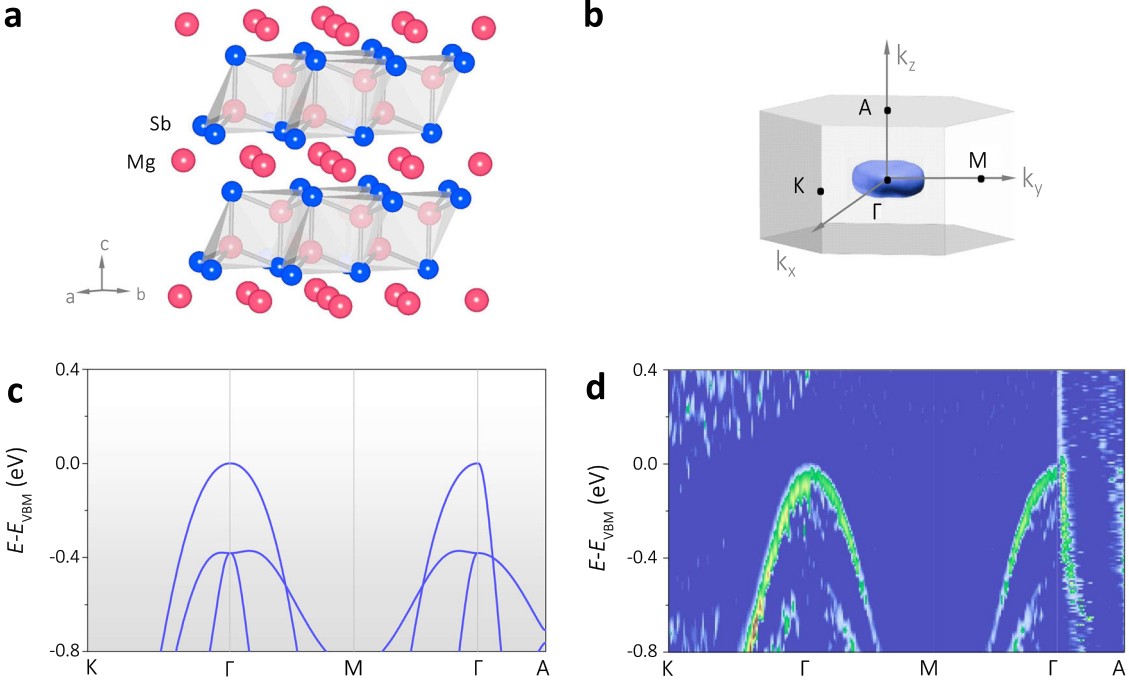

**Fig. 2 The crystal and band structures of Mg₃Sb₂. a** Crystal structure. **b** Calculated Fermi surface at a hole concentration of ~$10^{20}$ cm$^{-3}$, and **c** the calculated valence band structure for Mg₃Sb₂. **d** ARPES results of the valence band structure of Mg₃Sb₂, Γ−K and Γ−M show nearly the same curvature, while Γ−M and Γ−A show a distinct anisotropy, which corresponds to the plate-like Fermi surface presented in **b**.

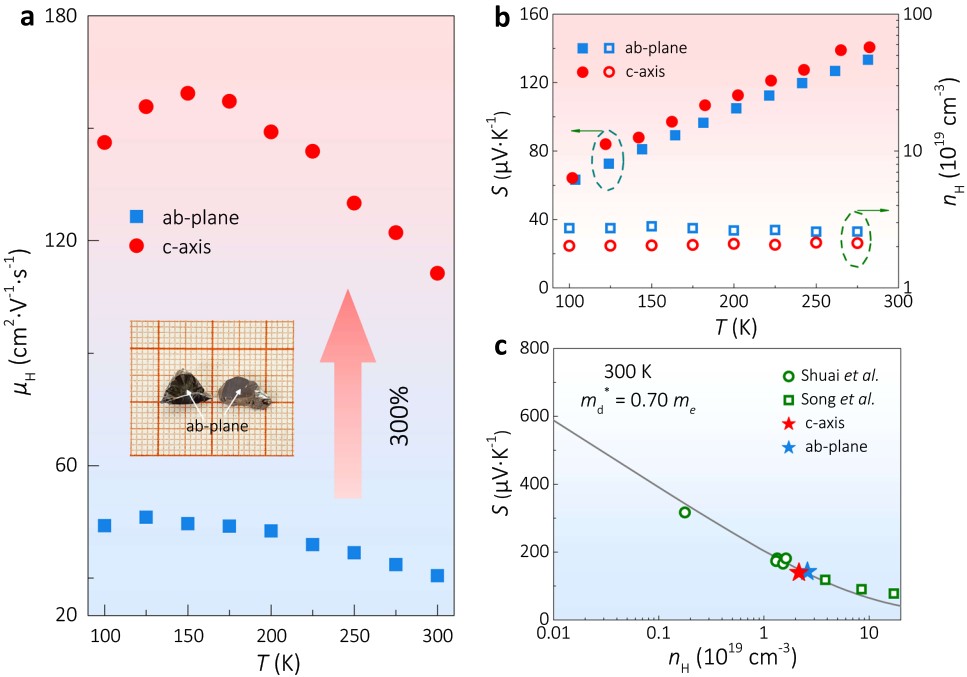

**Fig. 3 The electrical properties of single crystal Mg₃Sb₂.** Temperature dependences of **a** $\mu_H$, and **b** $S$ and $n_H$ for the Ag-doped Mg₃Sb₂ single crystal along c-axis and ab-plane. **c** Pisarenko plot showing the relationship between $S$ and $n_H$. The curve was calculated using the SPB model[18] with the $m_d^*$ of $0.70m_e$. The experimental data of the single crystals in this work and polycrystalline samples[32,33] are displayed. The inset in **a** is the optical image of the as-grown single crystals with a clear cleavage plane.

For p-type Mg₃Sb₂, even though the effective masses along ab-plane and c-axis directions are different, the $S$ should in principle be similar due to the unchanged Fermi level, as expected from Eq. (1). The relationship among $S$, $m_d^*$, and $n_H$ could be presented by the so-called Pisarenko plot. As shown in Fig. 3c, the experimental data of the single crystals along both directions agree well with the calculated line and also the data for polycrystalline samples[32,33]. This further confirms that $S$ is not orientation-dependent in p-type Mg₃Sb₂. The almost unchanged $S$ and $n_H$ but largely distinct $\mu$ along different directions demonstrate that the valley anisotropy is an effective means to decouple the $S$ and $\sigma$ to realize a better PF in p-type Mg₃Sb₂.

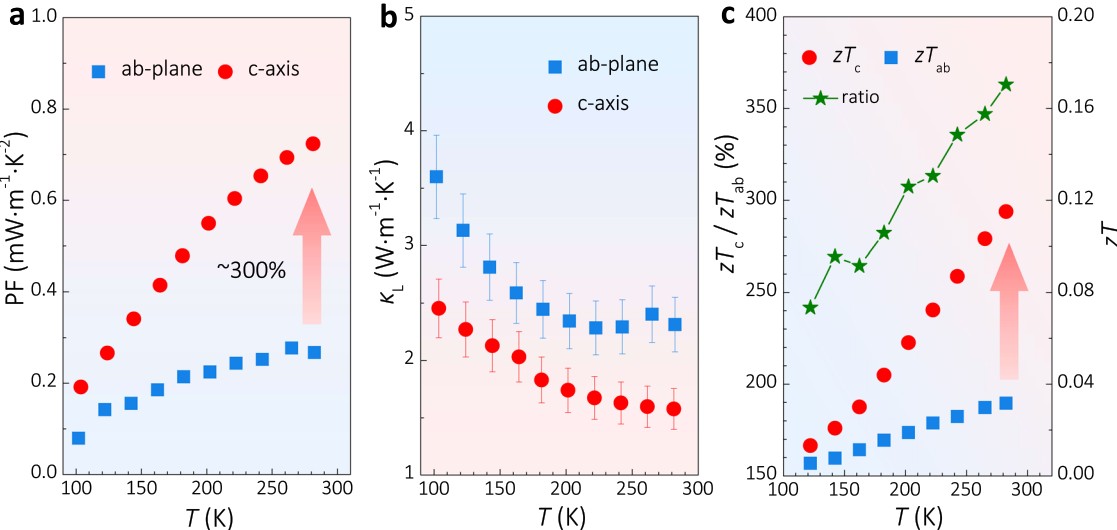

**Fig. 4 The TE properties of single-crystal Mg$_3$Sb$_2$.** Temperature dependences of **a** PF; **b** $\kappa_L$; **c** $zT$; and $zT$ ratio along c-axis and ab-plane for the Ag-doped Mg$_3$Sb$_2$ single crystal.

Because of the almost similar $S$ but much larger $\sigma$, the PF along the c-axis is nearly 300% of that along the ab-plane, as shown in Fig. 4a. We further measured the $\kappa$ along both directions using a four-probe steady-state method (schematically shown in Supplementary Fig. 3). The $\kappa_L$ was obtained by subtracting the electronic part of thermal conductivity according to the Wiedemann–Franz law. The results (Fig. 4b) imply a slight anisotropy in the $\kappa_L$ of Mg$_3$Sb$_2$, in which the $\kappa_L$ along c-axis is 1.6 W m$^{-1}$ K$^{-1}$ at 300 K, which is about 30% lower than that in ab-plane (2.3 W m$^{-1}$ K$^{-1}$ at 300 K). The slight upturn of $\kappa_L$ above 200 K might come from the heat radiation effect. Previously, Zhang et al. [34] argued that the $\kappa_L$ is nearly isotropic ($\kappa_a/\kappa_c \approx 1.1$ at 300 K) in Mg$_3$Sb$_2$ based on analyzing the chemical bonding in both intra-layer and inter-layer. Later on, the calculations using the finite temperature method by Zhu et al. [35] showed that there is an anisotropy in the $\kappa_L$ of Mg$_3$Sb$_2$. That is, the $\kappa_L$ along the c-axis is about 20% lower than that in the ab-plane. Together with current experimental results, we could conclude that the $\kappa_L$ of Mg$_3$Sb$_2$ along the c-axis is lower than that along the ab-plane, albeit slightly. One of the reasons leading to the anisotropy in $\kappa_L$ is that the average group velocities in the c-axis are smaller than the ab-plane, according to the calculations by Zhu et al. [35]. Besides, Mg1–Sb chemical bonding in the interlayer of Mg$_3$Sb$_2$ is weaker[34], which could be another factor leading to the anisotropy in $\kappa_L$.

Due to the much higher PF and smaller $\kappa_L$, Mg$_3$Sb$_2$ single crystal exhibits a larger $zT$ when the carrier transport along the c-axis (Fig. 4c). Specifically, near 300 K, the $zT$ along the c-axis is about 260% higher than that along the ab-plane. This result demonstrates a feasible strategy to realize a better TE performance by utilizing valley anisotropy.

**Textured polycrystalline Mg$_3$Sb$_2$.** Owing to the relatively small size of the as-grown Mg$_3$Sb$_2$ single crystal, the measurements of TE properties above room temperature are difficult using commercial equipment which usually requires samples in a centimeter size. Thus, to study the effect of valley anisotropy on high-temperature transport properties of Mg$_3$Sb$_2$, we further prepared the textured polycrystalline samples with suitable sizes (details shown in the method section), facilitating the measurements above room temperature. Figure 5a and Supplementary Fig. 4 show the XRD patterns of the textured polycrystalline samples along parallel (//P) and vertical (⊥P) to the pressing direction. The intensity ratio $I_{002}/I_{110}$ of ⊥P is 1.56, much stronger than that

of //P (1.05), indicating that the grains in the polycrystalline samples are textured. Moreover, the scanning electron microscopy (SEM) images of both //P and ⊥P are displayed in Supplementary Fig. 5, further verifying the textured structures.

Even if the samples are textured, similar $S$ along different directions is still observed (Fig. 5b). More importantly, the higher $\sigma$ of //P is higher (Fig. 5c), as expected from the single-crystal study (Fig. 3). Meanwhile, the $\kappa_L$ of the textured Mg$_3$Sb$_2$ polycrystalline samples also shows a slight anisotropic character (Fig. 5d), although it is weaker compared to that in the single crystal (Fig. 4b). Hence, the PF and $zT$ for the sample //P are much higher than that of ⊥P (Fig. 5e). Moreover, it is worth noting that the $zT$ of //P is improved in the whole temperature range. This indicates a higher device $ZT$[36] for the sample //P, which is about 60% higher than that of ⊥P (inset of Fig. 5e). As a supplemental confirmation, we also found a higher TE performance along with //P in other Mg$_{3-x}$Ag$_x$Sb$_{2-y}$Bi$_y$ samples, despite having different contents of Bi and Ag (Supplementary Figs. 6–9). The results from the textured polycrystalline samples, as well as the single crystals, demonstrate the effectiveness of utilizing valley anisotropy to enhance the TE performance.

Furthermore, it is meaningful to make a comparison of the anisotropic transport properties of the studied Mg$_3$Sb$_2$ with the other "layer-structured" TE materials, such as Bi$_2$Te$_3$, SnSe, and SnSe$_2$. Although all of them show anisotropic TE properties, the observed highest $zT$ occurs in different crystallographic directions, either the in-plane or the cross-plane direction. Both n-type and p-type Bi$_2$Te$_3$-based materials exhibit higher $zT$ along the in-plane direction, owing to the higher $\mu$[37], despite also having a higher $\kappa_L$ in the same direction. The n-type SnSe[38] and SnSe$_2$[39] show a higher $zT$ along the cross-plane direction, mainly contributed by the much lower $\kappa_L$. In contrast, p-type Mg$_3$Sb$_2$ is an exceptional case because that it shows not only a lower $\kappa_L$ but also a much higher $\mu$ in the cross-plane direction. It might be common for the "layer-structured" materials to show a lower $\kappa_L$ along the cross-plane direction but is rather rare to exhibit a much higher $\mu$. Therefore, a further understanding of the origin of the valley anisotropy in p-type Mg$_3$Sb$_2$ is important.

**Valley anisotropy in $AB_2X_2$ Zintl phase compounds.** The calculations of the orbital-projected band structures were performed and the enlarged details of the valence band maximum (VBM) for Mg$_3$Sb$_2$ are displayed in Fig. 6a. It is found that the $p_z$ orbital of

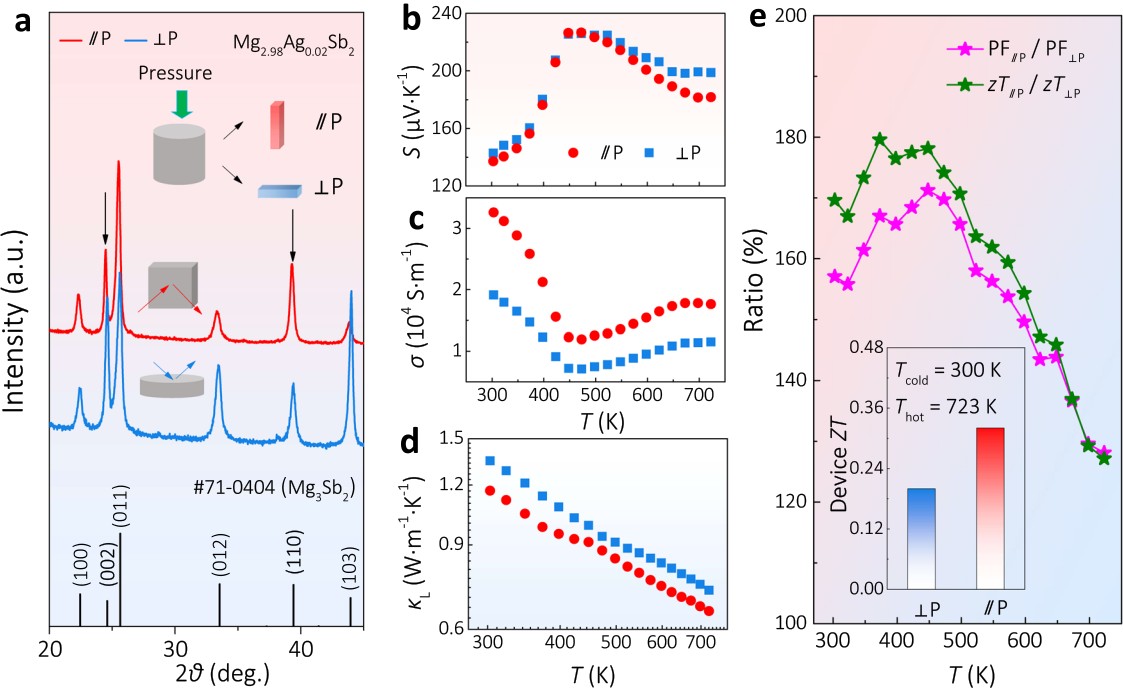

**Fig. 5 The structure and TE properties of textured polycrystalline Mg₃Sb₂.** **a** XRD patterns, temperature dependences of **b** $S$, **c** $\sigma$, **d** $\kappa_L$, **e** $zT$, and PF ratios for the textured polycrystalline Mg₃Sb₂ samples. The inset in **e** shows the calculated device $zT$[36].

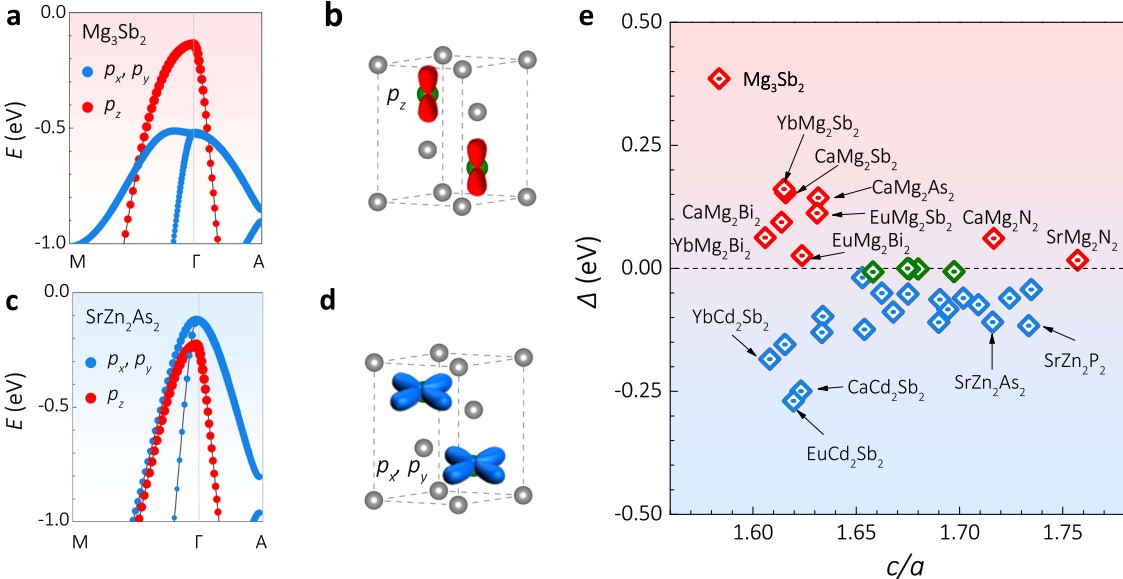

**Fig. 6 The band structure of $AB_2X_2$ Zintl phase compounds.** **a** Orbital projected band structures of Mg₃Sb₂. **b** Schematic diagram of the $p_z$ orbital-dominated hole transport in Mg₃Sb₂. **c** Orbital projected band structures of SrZn₂As₂. **d** Schematic diagram about ($p_x$, $p_y$) orbitals-dominated hole transport in SrZn₂As₂. **e** Crystal field splitting energy versus lattice ratio $c/a$ for $AB_2X_2$ Zintl phase system.

Sb dominates the VBM (red dots) at the Γ point. In real space, the dumbbell-shaped $p_z$ orbital corresponds to the anisotropy of the valence band (Fig. 6b). More specifically, $p_z$ orbital will be more overlapping along the c-axis than in the ab-plane, indicating that the valence band is much dispersive along the c-axis. The $p_z$-orbital dominated VBM phenomenon is a result of the crystal field splitting effect[40,41]. Mg₃Sb₂ belongs to the $AB_2X_2$ Zintl phase compounds family ($A$ is alkaline earth or lanthanides element, $B$ is Mg or VIIB or IIB or IIIA group element, $X$ is IVA or VA group element)[42], it is thus interesting to investigate whether other candidates from this family show a similar anisotropic

VBM. Owing to the effect of crystal field splitting, the position of $p_z$ orbital could be either above ($\Delta > 0$) or below ($\Delta < 0$) the ($p_x$, $p_y$) orbitals depending on the value of crystal field splitting energy $\Delta$ [$\Delta = E(p_z) - E(p_x, p_y)$]. The ($p_x$, $p_y$) orbitals-dominated VBM (blue dots in Fig. 6c) is observed in another Zintl compound SrZn₂As₂. Different from Mg₃Sb₂, more overlapping of ($p_x$, $p_y$) orbitals are expected in ab-plane (Fig. 6d) in real space, which implies higher hole mobility in the ab-plane for SrZn₂As₂.

The $AB_2X_2$ Zintl phase compounds are a big family which shows great potential to be explored as the TE candidates[43,44]. According to the Zintl chemistry, smaller $\kappa_L$ along the c-axis is

expected for all the $AB_2X_2$ compounds due to the ionic bonding between the interlayers[34,45]. If these compounds possess a similar anisotropic valence band structure to that of $Mg_3Sb_2$, it will promise higher TE performance when the carriers transport along the c-axis. To search for other potential candidates, the first-principles calculations were further conducted to investigate the band structure of 66 $AB_2X_2$ Zintl compounds (Supplementary Figs. 10–23, Supplementary Table 1). Out of them, we found 9 candidates (Fig. S9) that have a similar valence band anisotropy as $Mg_3Sb_2$. Moreover, the anisotropic factor $K$ was also estimated for the 9 candidates (Supplementary Table 2) and some of them are even larger than that of $Mg_3Sb_2$, indicating that they will also exhibit good TE performance along the c-axis. For example, $CaMg_2Bi_2$ is one of the 9 candidates. Previously, the $CaMg_2Bi_2$ polycrystalline sample, without considering the anisotropy, was found to be a good p-type TE material with a peak $zT$ of about 1.3 at 873 K[46]. According to our calculations, $CaMg_2Bi_2$ exhibits a strong valence band anisotropy ($K \sim 16$). Hence, if the single crystals or textured polycrystalline samples of $CaMg_2Bi_2$ can be made, one can expect even higher $zT$ along the c-axis owing to the higher $\mu$ and lower $\kappa_L$.

We also notice that some other $AB_2X_2$ compounds have a similar band structure as that of $SrZn_2As_2$ (Supplementary Figs. 13–16), indicating a higher $\mu$ along the ab-plane. The anisotropic TE performance of these $AB_2X_2$ compounds with ($p_x$, $p_y$)-orbitals-dominated VBM is thus determined by the trade-off between $\mu$ and $\kappa_L$ along the c-axis and ab-plane. It is worth noting that the crystal field splitting energy $\Delta$ of some $AB_2X_2$ compounds is nearly zero (green dots in Fig. 6e), indicating a high orbital band degeneracy. Under this condition, the anisotropy of VBM is not obvious, but the high orbital band degeneracy might benefit the improved TE performance, as previously argued by Zhang et al. [41,47] Thus, both band anisotropy and orbital degeneracy play significant roles in obtaining high TE performance in the light of their valence band structure. Different band engineering strategies, either band anisotropy or band convergence, can thus be considered to enhance their TE performance. Beyond $AB_2X_2$ Zintl phase compounds, recent years have witnessed the discoveries of many TE materials with non-cubic crystal structures. Hence the anisotropy in the electronic valley and phonon dispersion offers an additional degree of freedom that can be utilized to enhance TE performance.

## Discussion

From the above experiments on p-type $Mg_3Sb_2$, the facts why valley anisotropy can be utilized to realize the enhanced TE performance are summarized: one is the higher $\mu$ in the light-band direction, the other is the almost unchanged $S$ and $n$ in both light-band and heavy-band directions. These results suggest that the $\mu$ is more sensitive to the band curvature while $S$ and $n$ are not[48]. The anisotropic factor $K$ for p-type $Mg_3Sb_2$ is about 5.6, which is similar to other good TE materials, such as ZrNiSn ($K \sim 8$)[49], $Bi_2Te_3$ ($K \sim 2.6$)[50]. This indicates the generality of utilizing valley anisotropy to realize the enhanced PF in conventional TE materials. In some other cases, such as the layered metal $NaSn_2As_2$[51], the electronic structure shows multiple electron and hole pockets near the Fermi level. As a result, the $S$ of $NaSn_2As_2$ exhibits opposite signs along with different crystallographic directions, owing to the significant change in the Fermi surface curvature. Such cases are out of the scope of the current TE research focusing on semiconductors with a single type of carrier.

In summary, in our opinion, the $S$ will not display an obvious difference along with different crystallographic directions for most semiconductors with valley anisotropy. However, $\mu$ is very sensitive to the curvature of the electronic valley, which offers an

opportunity to realize higher $\mu$ while keeping $S$ unchanged. This is different from the previous band engineering strategies, such as band convergence and resonant states, which focus on the enhancement of the $S$ by regulating the DOS. Taking p-type $Mg_3Sb_2$ as a model system, we demonstrate that valley anisotropy can be utilized as an effective means to decouple the correlation between $S$ and $\mu$, leading to the realization of enhanced power factor. Compared to the heavy-band direction, the carriers show much higher $\mu$ while the $S$ keeps unchanged if they transport along the light-band direction, resulting in an increased power factor by a factor of 3. This conclusion is also suitable to other isostructural $AB_2X_2$ Zintl phase compounds with $p_z$ orbital-dominated valance band, which usually have a lower $\kappa_L$ along the light-band direction as well. Moreover, utilizing valley anisotropy can also be a feasible strategy to enhance the performance of other TE materials with anisotropic electronic structures.

## Methods

**Sample preparation**. For synthesizing the single-crystal samples, starting elements Mg (granules, 99.8%), Sb (shot, 99.999%), and Ag (shot, 99.999%) were weighted and mixed with a molar ratio of Mg:Sb:Ag = 2.98:2:0.02. The tantalum tubes with an inner diameter of 10 mm were used to store the mixtures and then sealed under argon atmosphere using an arc melter. Afterward, these tantalum tubes were put into the sealed quartz tubes and heated up to 1453 K, and dwelled for 24 h ensuring homogeneity. For crystal growth, the tubes were slowly cooled down to 923 K with a cooling rate of 2.5 K/h. For preparing the polycrystalline samples $Mg_{3-x}Ag_xSb_2$ and $Mg_{3-x}Ag_xSb_{1.5}Bi_{0.5}$ ($x = 0.015, 0.02, 0.025$), starting elements Mg (granules, 99.8%), Sb (shot, 99.999%), Bi (shot, 99.999%), and Ag (shot, 99.999%) were weighed and mixed nominally and then were melted in Ta tube as the same procedure as described for single crystals. The obtained ingots ($\approx 12$ g) were loaded into graphite dies (diameter: 12.7 mm) for spark plasma sintering process (SPS) (SPS-1050; Sumitomo Coal Mining Co.) under 923 K and 60 MPa in a vacuum for 10 min. For structural characterization and transport measurements, the obtained samples were cut along the directions that are parallel and perpendicular to the pressing direction, respectively, as shown in the inset of Fig. 5a.

**Characterization**. The single crystals were checked and oriented at room temperature by the white-beam backscattering Laue X-ray diffraction (XRD) method (Rigaku AFC7 plus Saturn 724 + CCD). For the polycrystalline samples, the XRD (Rigaku, Rint 2000, CuK$_\alpha$) and SEM (Hitachi, S-3400N, 15 kV) analyses were performed along parallel (//P) and vertical ($\perp$P) to the pressing direction to characterize the texture. To experimentally reveal the valence band structure, we carried out the angle-resolved photoemission spectroscopy (ARPES) measurements of $Mg_3Sb_2$ single crystals at the UE112-PGM2b beamline of the synchrotron radiation facility BESSY (Berlin) with $1^3$ and $1^2$ end stations, equipped with DA30L and R800 analyzers, respectively. The total energy resolution is about 15 meV while the angular resolution is 0.2°. The crystals were cleaved in situ and measured at 20 K.

**Measurements**. Single crystals with dimensions of about $1 \times 1 \times 3$ mm$^3$ were employed for the transport measurements. The resistivity ($\rho$) and Hall resistivity ($\rho_H$) were measured simultaneously using a PPMS-9T instrument (Quantum Design) using the ACT option via a standard four-probe method. Hall carrier density $n_H$ and mobility $\mu_H$ are obtained from $n_H = 1/eR_H$, and $\mu_H = R_H/\rho$, where $R_H$ is the Hall coefficient. TE transport properties were measured with a steady-state heat sink method in the high vacuum condition with a breakout box[51]. A strain gauge heater was placed on one end of the sample to apply the heat power. Two type-E thermocouples were attached along with the sample for measuring the temperature difference, with the chromel leg used for thermopower measurement. The thermal conductivity $\kappa$ was then calculated by the formula $\kappa = Wl/(\Delta Tdw)$, where $W$ is the applied heater power, $\Delta T$ is the temperature difference, $l$, $d$, and $w$ are the length between the two thermocouples, width, and thickness of the sample, respectively. For measuring the $S$, we applied various heater currents at selected temperatures and measured the TE voltage as a function of temperature difference. The slope of $\Delta V$ over $\Delta T$ is taken as the relative Seebeck with respect to the chromel. For the polycrystalline sample, the commercial Linseis LSR-3 system was used for the measurements of the Seebeck coefficient $S$ and the electrical conductivity $\sigma$ from 300 to 700 K with an accuracy of about $\pm 5\%$ and $\pm 3\%$, respectively. Netzsch LFA457 was used for the measurement of the thermal diffusivity $D$ with an accuracy of about $\pm 3\%$. The thermal conductivity $\kappa$ was calculated from the equation: $\kappa = D\rho C_P$. The sample density $\rho$ was estimated by the Archimedes method and the heat capacity $C_P$ was calculated by the method proposed by Agne et al. [52]: $C_P = 3NR/M_w \times (1 + 1.3 \times 10^{-4}T - 4 \times 10^3T^{-2})$, where $N$ is the number of elements in the formula unit, $R$ is the gas constant, $M_w$ is the molecular weight of the formula unit.

**Band structure calculations**. The density functional theory was employed in this work using the Vienna ab initio Simulation Package[53,54] with the projector augmented-wave method[55]. Perdew–Burke–Ernzerhof type generalized gradient approximation was used as the exchange-correlation functional[56]. A plane-wave energy cutoff of 400 eV and an energy convergence criterion of $10^{-5}$ eV for self-consistency was adopted. All the atomic positions were relaxed to equilibrium until the calculated Hellmann–Feynman force on each atom was <0.01 eV/Å. The Monkhorst–Pack uniform $k$-point sampling with $k = 60/L$ ($L$ is the corresponding lattice parameter) was used in self-consistent static calculations for charge density and dielectric constant. Non-self-consistent calculations were then performed to calculate the band structures using the converged charge density and a Gaussian broadening of 0.05 eV was used.

**Reporting summary**. Further information on experimental design is available in the Nature Research Reporting Summary linked to this paper.

## Data availability

The experiment data that support the findings of this study are available from the corresponding author upon reasonable request.

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

## Acknowledgements
This work was supported by the National Key Research and Development Program of China (2019YFA0704902), the National Science Fund for Distinguished Young Scholars (No. 51725102), the National Natural Science Foundation of China (Nos. 52101275, 51761135127), and the Deutsche Forschungsgemeinschaft (DFG, German Research Foundation)—Projektnummer (392228380).

## Author contributions
A.L., C. Fu, and T.Z. designed the project. A.L. prepared the samples and carried out the transport measurements. A.L. characterized the crystals with input from Y.W. C.H. performed first-principles calculations. B.H. measured the thermal transport properties of single crystals. M.Y. carried out the angle-resolved photoemission spectroscopy study. X.Z. and C. Felser supervised the project. A.L. and C. Fu analyzed the data and wrote the original manuscript. All the authors reviewed and edited the manuscript.

## Competing interests
The authors declare no competing interests.
