## [Peer Review File · Nature Communications]

Editorial Note: Parts of this peer review file have been redacted as indicated to remove third-party material where no permission to publish could be obtained.

REVIEWER COMMENTS

Reviewer #1 (Remarks to the Author):

The present manuscript tried to prove the effectiveness of valley anisotropy to enhance the thermoelectric performance, where taking p-type Zintl phase compound Mg_3Sb_2 as an example, the enhanced power factor was obtained. Meanwhile, the electronic structures of 66 isostructural Zintl phase compounds were calculated, showing that 9 of them have the similar valence band anisotropy as Mg_3Sb_2 , but no directly experimental results are provided.

1. P 6: The authors claim that the effective mass of Γ -K and Γ -A is similar with that of reports of Zhang et al. and Meng et al, but the difference of $m_{\Gamma\text{-K}}$ is actually over 30%.
2. P7: "These results confirm that the carrier transport along the light-band direction is faster than that along the heavy-band direction owing to the larger carrier velocity", actually, i didn't find the direct evidence.
3. For the single crystal Mg_3Sb_2 , an anisotropy in κ_L is found, which is just the result of the difference in the average group velocities along different directions? If yes, please provide more details.
4. The texture itself is beneficial to the improvement in the electrical properties along c axis, where it is not precise to suggest the effectiveness of valley anisotropy to enhance the thermoelectric performance by taking the textured samples as an example.
5. By the first principles calculations, the authors calculated the band structures of 66 AB₂X₂ Zintl compounds, where 9 compounds among these materials show the similar valence band isotropy as Mg_3Sb_2 . Please provide more experimental details to prove that. This is also helpful for readers to understand this work.

Reviewer #2 (Remarks to the Author):

In the past years, thermoelectric materials research has experienced rapid development owing to the proposed new strategies. Band engineering, such as the resonant states and band convergence, has been the leading strategy that improves the electrical performance of thermoelectric materials. However, the simultaneous realization of large Seebeck coefficient and electrical conductivity is very challenging, as they have opposite dependences on the carrier concentration. In this work, Li et al demonstrate that valley anisotropy can be utilized to enhance the electrical conductivity but keep Seebeck coefficient unchanged. They found a simple material system, p-type Mg_3Sb_2 , to demonstrate the effect of valley anisotropy on electrical properties. The most important finding is that, compared to the heavy-band direction, they observed a 3-times higher electrical conductivity in the light-band direction of p-type Mg_3Sb_2 single crystal while the Seebeck coefficient remains similar. Moreover, they also demonstrated the efficacy of this strategy in the textured polycrystalline samples and predicted new candidates through first-principles calculations. This work is significant and original, the manuscript is well organized and written. I think it represents an important advance in the thermoelectric community and will inspire more experimental and also theoretical studies. I am happy to recommend to publish it in Nature Communications after some minor comments are addressed.

1) Understandably, the authors take p-type Mg_3Sb_2 for the current study since it has one hole pocket. How do the authors comment on the material systems that have multiple pockets? That is, how to utilize the proposed strategy to enhance the TE performance of materials with multiple pockets?

- 2) It makes sense that the Seebeck coefficient show nearly an isotropic property along different direction for the valley with relatively small anisotropy. How is the material that exhibits a large band anisotropy, Is it always true that the Seebeck coefficient is isotropic?
- 3) Figure 5 b and c, the Seebeck coefficient and electrical conductivity show a transition of the slope near 450 K, which however does happen to the lattice thermal conductivity, does it relate to the bipolar effect, or how to understand such a phenomenon?

Reviewer #3 (Remarks to the Author):

This work investigated the anisotropic thermoelectric transport properties of p-type single crystal and polycrystal Mg_3Sb_2 , revealing a difference in carrier mobility and electrical conductivity along ab-plane and c-axis directions. This can further be understood by the difference in inertial mass along these directions. The anisotropic valence band structure is well demonstrated through both calculation and ARPES measurements on Mg_3Sb_2 single crystal. I recommend the following revisions prior to a publication in NC.

1. Why Mg_3Sb_2 single crystal shows a difference in carrier concentration along ab-plane and along c-axis? The homogeneity in chemical composition needs to be confirmed.
2. Increased lattice thermal conductivity at $T > 200$ K for Mg_3Sb_2 single crystal along the ab-plane is abnormal, more discussion should be included.
3. Original Laue diffraction results should be included in manuscript. In addition, XRD measurements is suggested to ensure the quality of crystal orientation of the bulk crystal.
4. Mg_3Sb_2 has the largest crystal field splitting energy among 1-2-2 Zintl compounds, which is not desired for thermoelectric performance due to the low band degeneracy. The expression of anisotropy here is a bit of misleading. Ideally, isotropic and low inertial effective mass, rather than anisotropy itself, is the key factor for a high power factor along all directions. The importance of anisotropic valley should not be overemphasized, since anisotropic thermoelectric performance have strong drawbacks for practical applications.
5. To better understand the effect of band anisotropy and crystal field splitting energy on electronic performance, a detailed comparison on experimental power factor for 1-2-2 Zintls can be helpful.

Response to Reviewers

Reviewer #1:

Comment: The present manuscript tried to prove the effectiveness of valley anisotropy to enhance the thermoelectric performance, where taking p-type Zintl phase compound Mg₃Sb₂ as an example, the enhanced power factor was obtained. Meanwhile, the electronic structures of 66 isostructural Zintl phase compounds were calculated, showing that 9 of them have the similar valence band anisotropy as Mg₃Sb₂, but no directly experimental results are provided.

Response: Thanks for your careful reading and comment. In this work, we take p-type Mg₃Sb₂ as an example to demonstrate the effectiveness of valley anisotropy to enhance the thermoelectric performance because of the clarity of its anisotropic valence band and the availability of single crystals and textured polycrystals. To give more experimental evidence, we have supplemented an additional experiment on CaMg₂Sb₂, selected from the predicted 9 candidates. The results support the conclusions that we draw from the studies of Mg₃Sb₂. Detailed experimental results on CaMg₂Sb₂ (Figure R1) are presented below as the reply to Comment 5.

Comment: 1. P6: The authors claim that the effective mass of Γ -K and Γ -A is similar with that of reports of Zhang et al. and Meng et al, but the difference of $m_{\Gamma-K}^*$ is actually over 30%.

Response: Thanks for your comment. The effective mass is derived from the calculated band structure, which depends on the details of the calculated band structure. Zhang et al. (Wien2k), Meng et al. (VASP + LDA), and we (VASP + PBE) adopt three different calculation methods, which could generate different band structures. Although there is a difference in the effective mass among different calculations, all the results indeed show the existence of valence band anisotropy in Mg₃Sb₂, which is a key point in this work. To avoid possible confusion, we revised the sentence in P6 to: "... A K value of about 8.2 is thus derived. This result is close to the calculations by Zhang *et al.*³¹ ($m_{\Gamma-K}^* = 1.15 m_e$, $m_{\Gamma-A}^* = 0.15 m_e$) and Meng *et al.*³⁰ ($m_{\Gamma-K}^* = 0.61 m_e$, $m_{\Gamma-A}^* = 0.07 m_e$), giving the K values of 7.7 and 8.7, respectively."

Comment: 2. P7: "These results confirm that the carrier transport along the light-band direction is faster than that along the heavy-band direction owing to the larger carrier

velocity”, actually, i didn’t find the direct evidence.

Response: Thanks for your comment. Experimentally, we observed much higher carrier mobility along the light-band direction than that along the heavy-band direction. This can be a result of the weaker carrier scattering, the lower inertial effective mass (indicating a larger carrier velocity), or both. To be more rigorous and to avoid possible confusion, we revised this expression in P7 to: “These results suggest that the carrier transport along the light-band direction is faster than that along the heavy-band direction, probably owing to the weaker carrier scattering and low inertial effective mass.”

Comment: 3. For the single crystal Mg₃Sb₂, an anisotropy in κ_L is found, which is just the result of the difference in the average group velocities along different directions? If yes, please provide more details.

Response: Thanks for this question. In the previous calculations by Zhu et al. (Ref. 35), they pointed out that the calculated average group velocities in the c-axis are smaller than the a-axis. Thus, we think the difference in average group velocities should be one of the reasons that lead to the anisotropy in κ_L . Additionally, as motivated by the reviewer’s comment, we think there should be other reasons contributing to the anisotropy in κ_L . The anisotropy can also be traced back into their chemical bonding. The weaker Mg1-Sb ionic bonds exist in the interlayer of Mg₃Sb₂. This may also bring about the difference in the phonon scattering along different directions. Therefore, we add more discussions about the origin of the anisotropy of κ_L in Mg₃Sb₂ in P9: “One of the reasons leading to the anisotropy in κ_L of is the average group velocities in the c-axis are smaller than the ab-plane, as from the calculations by Zhu *et al.*³⁵. Besides, Mg1-Sb chemical bonding in the interlayer of Mg₃Sb₂ is weaker³⁴, which could be another reason leading to t the anisotropy in κ_L .”

Comment: 4. The texture itself is beneficial to the improvement in the electrical properties along c axis, where it is not precise to suggest the effectiveness of valley anisotropy to enhance the thermoelectric performance by taking the textured samples as an example.

Response: Thanks for your comment. The effectiveness of valley anisotropy to enhance the thermoelectric performance is justified by performing the Mg₃Sb₂ single-crystal study. For the textured polycrystals study, we would like to elaborate that the enhancement of texture of the polycrystalline samples can be a means to improve the electrical properties in case large

single crystals are difficult to synthesize. Given this, we think the valley anisotropy is one of the main reasons leading to the improved electrical properties in the textured polycrystals (the out-plane direction for Mg_3Sb_2). To avoid confusion, we revised some expressions in the related paragraphs.

Comment: 5. By the first principles calculations, the authors calculated the band structures of 66 AB_2X_2 Zintl compounds, where 9 compounds among these materials show the similar valence band isotropy as Mg_3Sb_2 . Please provide more experimental details to prove that. This is also helpful for readers to understand this work.

Response: Thank you for your suggestions. We have supplemented additional experimental results on CaMg_2Sb_2 selected from our predictions. Since CaMg_2Sb_2 has a higher melting point than that of Mg_3Sb_2 , we did not obtain a large single crystal but succeed in preparing the oriented polycrystalline CaMg_2Sb_2 . As presented in Figure R1, the XRD patterns suggest that sample S2 has more grains oriented along the c-axis than S1. The Seebeck coefficient of the two samples is similar, but the electrical conductivity is higher for S2. We think valley anisotropy contributes to the improved electrical properties in S2, as what we have found in the study of the textured polycrystalline Mg_3Sb_2 . Thus, we believe valley anisotropy provides a means to enhance the TE performance of the predicted AB_2X_2 Zintl phase compounds. This could motivate more experimental studies on this aspect.

[Redacted]

Figure R1. The XRD patterns and electrical properties of the two CaMg_2Sb_2 polycrystalline samples, in which S2 is a sample having more grains oriented along the c-axis direction than S1.

Reviewer #2:

Comment: In the past years, thermoelectric materials research has experienced rapid development owing to the proposed new strategies. Band engineering, such as the resonant states and band convergence, has been the leading strategy that improves the electrical performance of thermoelectric materials. However, the simultaneous realization of large Seebeck coefficient and electrical conductivity is very challenging, as they have opposite dependences on the carrier concentration. In this work, Li et al demonstrate that valley anisotropy can be utilized to enhance the electrical conductivity but keep Seebeck coefficient unchanged. They found a simple material system, p-type Mg_3Sb_2 , to demonstrate the effect of valley anisotropy on electrical properties. The most important finding is that, compared to the heavy-band direction, they observed a 3-times higher electrical conductivity in the light-band direction of p-type Mg_3Sb_2 single crystal while the Seebeck coefficient remains similar.

Moreover, they also demonstrated the efficacy of this strategy in the textured polycrystalline samples and predicted new candidates through first-principles calculations. This work is significant and original, the manuscript is well organized and written. I think it represents an important advance in the thermoelectric community and will inspire more experimental and also theoretical studies. I am happy to recommend to publish it in Nature Communications after some minor comments are addressed.

Response: Thank you very much for the positive comments on our work.

Comment: 1) Understandably, the authors take p-type Mg_3Sb_2 for the current study since it has one hole pocket. How do the authors comment on the material systems that have multiple pockets? That is, how to utilize the proposed strategy to enhance the TE performance of materials with multiple pockets?

Response: Thanks for your insightful questions. In the past, multiple pockets have long been considered as a good indicator for excellent thermoelectric materials, while the anisotropy has also been considered as good as for TE performance. However, there is no experimental work to solely study the band anisotropy. In this work, we choose a single-pocket material for study by focusing on its anisotropic character. This excludes the effect of band degeneracy on thermoelectric performance and largely simplifies analysis and discussion. Under this circumstance, we recognize the relations between S and μ and prove its advantages in thermoelectrics. We think the strategy proposed in this work is also suitable for materials with

multiple pockets. That is, if materials hold multiple pockets with anisotropy, a higher electrical power factor can still be expected when carrier transport in the light band direction.

Comment: 2) It makes sense that the Seebeck coefficient show nearly an isotropic property along different direction for the valley with relatively small anisotropy. How is the material that exhibits a large band anisotropy, Is it always true that the Seebeck coefficient is isotropic?

Response: Thanks for your questions. In the discussion part, we discussed the anisotropy of the Seebeck coefficient. The Seebeck coefficient is only the function of carrier density under the acoustic phonon scattering of carriers. However, when band anisotropy is very large, for example, with a K value > 100 , it might lead to different charge carrier scattering mechanisms along different directions. This might result in the anisotropic character of the Seebeck coefficient. Meanwhile, when there are two types of carriers (holes and electrons), it may even lead to a different sign of the Seebeck coefficient along different directions, for example, NaZn_2As_2 (ref. 51). Therefore, to our opinion, the Seebeck coefficient might exhibit an anisotropic character, but under the common circumstance that most good thermoelectric semiconductors follow (acoustic phonon scattering of carrier and only one kind of charge carriers), we think the Seebeck coefficient shall be similar in different crystallographic directions.

Comment: 3) Figure 5 b and c, the Seebeck coefficient and electrical conductivity show a transition of the slope near 450 K, which however does happen to the lattice thermal conductivity, does it relate to the bipolar effect, or how to understand such a phenomenon?

Response: Thank you very much for your careful reading and good questions. We have noticed this uncommon phenomenon. We do not believe this is caused by the bipolar effect because the lattice thermal conductivity does not provide evidence of the bipolar heat conduction. In our opinion, the transition near 450 K might be caused by Ag atoms, who might play as the “dynamic dopant”. In Ref. 32, Song et al. have measured the temperature dependence of carrier concentration in Ag-doped samples. An increasing carrier density has been found. We guess this might be related to the Ag atom, which has a small atomic and ionic size. They might be activated at this special temperature and provide the sudden increasing carrier density in the samples.

Reviewer #3:

Comment: This work investigated the anisotropic thermoelectric transport properties of p-type single crystal and polycrystal Mg₃Sb₂, revealing a difference in carrier mobility and electrical conductivity along ab-plane and c-axis directions. This can further be understood by the difference in inertial mass along these directions. The anisotropic valence band structure is well demonstrated through both calculation and ARPES measurements on Mg₃Sb₂ single crystal. I recommend the following revisions prior to a publication in NC.

Response: Thank you very much for your careful readings and suggestions.

Comment: 1. Why Mg₃Sb₂ single crystal shows a difference in carrier concentration along ab-plane and along c-axis? The homogeneity in chemical composition needs to be confirmed.

Response: Thanks for your questions. We think one possible reason for the difference in carrier concentration could be from the measurement uncertainty. Moreover, to check the homogeneity, the EDS mapping and line scanning along ab-plane and c-axis direction are performed for the Mg₃Sb₂ single crystal (See Supplementary Fig. 2). The results suggest that the chemical composition is homogeneous.

Comment: 2. Increased lattice thermal conductivity at T>200 K for Mg₃Sb₂ single crystal along the ab-plane is abnormal, more discussion should be included.

Response: Thanks for your comment. We think the abnormal lattice thermal conductivity at the high temperature mainly comes from measurement errors. In this work, we take the stationary method to measure the low-temperature thermal conductivity. A high vacuum is required to reduce the heat convection. However, at around room temperature, another way of heat conduction, heat radiation, becomes important. In the figure, we added the error bars for the measurement. Meanwhile, to avoid confusion, we add a short discussion about this in related paragraphs: “The slight upturn of κ_L when the temperature exceeds 200 K might come from the heat radiation effect, which becomes significant at high temperatures.”

Comment: 3. Original Laue diffraction results should be included in manuscript. In addition, XRD measurements is suggested to ensure the quality of crystal orientation of the bulk crystal.

Response: Thanks for your suggestion. Both original Laue diffraction results and XRD measurements have been provided in the supporting information of the manuscript (Supplementary Fig. 1).

Comment: 4. Mg₃Sb₂ has the largest crystal field splitting energy among 1-2-2 Zintl compounds, which is not desired for thermoelectric performance due to the low band degeneracy. The expression of anisotropy here is a bit of misleading. Ideally, isotropic and low inertial effective mass, rather than anisotropy itself, is the key factor for a high power factor along all directions. The importance of anisotropic valley should not be overemphasized, since anisotropic thermoelectric performance have strong drawbacks for practical applications.

Response: Thanks for the comment. We do not fully agree with the reviewer's arguments. First, the selection of p-type Mg₃Sb₂ for performing the current study is owing to its single Fermi pocket, which could serve as a good diagram to demonstrate the effect of band anisotropy on the TE performance due to its simplicity and cleanness. Second, we agree with the reviewer that high band degeneracy is beneficial for good thermoelectric performance. Third, we do not agree that an isotropic and low inertial effective mass is ideal for a high power factor. Rather than that, we think an anisotropic and low inertial effective mass is ideal for a high power factor, as demonstrated in this work. For two bands with the same m_b^* , the anisotropic one promises the same carrier density and Seebeck coefficient but high carrier mobility (owing to the lower inertial effective mass, as demonstrated in Figure R2), compared to the isotropic one. In all, we believe a material with multiple anisotropic pockets is the key factor for a high power factor.

[Redacted]

Figure R2. Band anisotropy dependence of effective masses when single-band effective mass remains constant. (*Adv. Mater.* 2017, 29, 1605884)

For practical applications, the question of whether the anisotropy is good or not is still waiting for more investigations. One example is the Bi₂Te₃-based system, which shows anisotropic

transport properties but is the most important commercialized thermoelectric system.

Comment: 5. To better understand the effect of band anisotropy and crystal field splitting energy on electronic performance, a detailed comparison on experimental power factor for 1-2-2 Zintl compounds can be helpful.

Response: Thanks for the comment. After a considered view, we think it is difficult to make a fair comparison on experimental power factor of 1-2-2 Zintl compounds, as the power factor depends significantly on the carrier concentration, the effective mass, the band degeneracy, the deformation potential and also the band anisotropy. Thus, we did not provide such a comparison. In this work, we do not emphasize the effect of crystal field splitting energy on electronic performance. We think the crystal field splitting energy can serve as an indicator to classify the 1-2-2 Zintl compounds into three categories. In the past, orbital engineering strategies have been proposed to enhance the band degeneracy for getting a higher performance. Here we focus on the compounds with the p_z -orbital-dominated single band, such as YbMg_2Sb_2 and CaMg_2Bi_2 , which were exploited as good thermoelectrics recently, their anisotropic band can be utilized to obtain higher TE performance as demonstrated in Mg_3Sb_2 . The past studies are just focused on polycrystalline samples, and their anisotropic bands have not been recognized to enhance the TE performance. Therefore, we think it is promising to expect a higher TE performance in these Zintls when considering their anisotropic bands. In all, in AB_2X_2 Zintl compounds, both orbital degeneracy and band anisotropic can be engineered for the enhancement of TE performance. Thus, we would expect that an ideal material system will be the one with the multiple anisotropic bands. We have added additional discussion in the related paragraphs.

REVIEWERS' COMMENTS

Reviewer #1 (Remarks to the Author):

The questions have been answered, and the revised manuscript is satisfactory for publication in Nature Communications.

Reviewer #2 (Remarks to the Author):

I think the manuscript is now ready to be accept. Congrates

Reviewer #3 (Remarks to the Author):

The revised manuscript seems now acceptable for this journal, I have no further comments.

Response to Reviewers

Reviewer #1:

Comment: The questions have been answered, and the revised manuscript is satisfactory for publication in Nature Communications.

Response: Thank you for the positive comment.

Reviewer #2

Comment: I think the manuscript is now ready to be accept. Congrates

Response: Thank you for the positive comment.

Reviewer #3

Comment: The revised manuscript seems now acceptable for this journal, I have no further comments.

Response: Thank you for the positive comment.